# Elevated Serum CCL23 Levels at Admission Predict Delayed Cerebral Ischemia and Functional Outcome after Aneurysmal Subarachnoid Hemorrhage

**DOI:** 10.3390/jcm11236879

**Published:** 2022-11-22

**Authors:** Hongwei Lin, Jie Shen, Yu Zhu, Lihui Zhou, Fan Wu, Zongchi Liu, Shengxiang Zhang, Renya Zhan

**Affiliations:** Department of Neurosurgery, The First Affiliated Hospital, School of Medicine, Zhejiang University, Hangzhou 310003, China

**Keywords:** CCL23, aneurysmal subarachnoid hemorrhage, functional outcome, neutrophil, delayed cerebral ischemia

## Abstract

(1) Background: CC chemokine ligand 23 (CCL23) is a chemokine implicated in the inflammatory response following brain damage. The aim of this study is to identify the change in serum CCL23 levels within 24 h after aSAH and whether serum CCL23 levels are associated with initial clinical severity, delayed cerebral ischemia (DCI), and functional outcome in patients with aneurysmal subarachnoid hemorrhage (aSAH). (2) Methods: 102 patients with aSAH and 61 controls were included in this prospective observational study. All clinical data were collected prospectively, and their serum CCL23 levels were measured. Initial clinical severity was reflected by the Hunt–Hess score and mFisher score. Functional outcome was evaluated in terms of the Glasgow Outcome Scale (GOS) score at 6-month follow-up. (3) Results: Patients with aSAH had higher serum CCL23 levels than controls. The temporal profile of serum CCL23 levels and neutrophils count exhibited a sustained increase within 24 h after aSAH. Serum CCL23 levels were related to blood neutrophils count, blood CRP levels, and initial clinical severity. Serum CCL23 level was an independent predictor of DCI and 6-month poor outcome in aSAH patients. (4) Conclusions: Serum CCL23 levels emerged as an independent predictor for DCI and poor outcome in patients with aSAH.

## 1. Introduction

Aneurysmal subarachnoid hemorrhage (aSAH) is an uncommon but life-threatening cerebrovascular disease worldwide. Despite substantial advances in the diagnosis and treatment of aSAH, its mortality and morbidity remain unfortunately high [1,2,3]. Approximately 41.7% of patients died within 28 days after SAH ictus, and 46% of survivors have long-term sequelae, including neuromotor disability and cognitive impairment [2,4,5].

The initial bleeding and delayed ischemia following aSAH have a major impact on the high disability rate and mortality of aSAH [6]. Initial clinical severity and delayed cerebral ischemia (DCI) [7,8,9] have proven to be the most powerful predictors of clinical outcome. Rapid identification and treatment of initial clinical severity and DCI contribute to alleviating permanent brain injury and improving the prognosis of aSAH patients [10,11,12]. Therefore, identification of a robust biomarker for evaluating initial clinical severity and predicting DCI would be a major step forward in the management of aSAH.

Immune reaction is increasingly recognized to have a pronounced impact on early brain injury and DCI following aSAH [13,14]. Chemokines are thought to be the most potent molecules that modulate immune cell trafficking and recruitment in the inflammatory responses [15,16]. CC chemokine ligand 23 (CCL23), a member of the CC chemokine family, is secreted by several immune cells (e.g., neutrophils, monocytes, eosinophils, dendritic cells, and macrophages) [17,18,19,20,21]. By specifically binding to CC chemokine receptor 1 (CCR1), secreted CCL23 exerts an impact on the production of pro-inflammatory cytokines and adhesion molecules that contributes to the migration of circulating immune cells to the inflamed environment [22,23].

Serum CCL23 has been implicated in multiple inflammatory diseases, such as systemic sclerosis, chronic rhinosinusitis, chronic kidney disease, and atherosclerotic disease [18,24,25,26]. Notably, CCL23 is also involved in the inflammatory response of brain injury and serves as a prognostic biomarker of Alzheimer’s disease and cerebral ischemia [27,28,29]. Here, we proposed that CCL23 might have a role in the inflammation of aSAH. In this study, we investigated whether serum CCL23 levels were specifically elevated in ICH patients compared to healthy controls. Then, we sought to identify whether serum CCL23 levels were related to inflammation, initial clinical severity, DCI, and functional outcomes in patients with aSAH.

## 2. Patients and Methods

### 2.1. Participants and Study Design

This was a prospective observational study of aSAH patients admitted to the Department of Neurosurgery at the First Affiliated Hospital, School of Medicine, Zhejiang University, between April 2017 and June 2019. The inclusion criteria for the study were (1) age ≥ 18 years; (2) SAH due to a ruptured aneurysm confirmed by computerized tomography (CT) angiography with or without digital subtraction angiography (DSA); (3) first-ever SAH; and (4) symptom onset within the preceding 24 h. Exclusion criteria were (1) SAH from other secondary causes (arteriovenous malformation (AVM), trauma, brain tumors, and hemorrhagic transformation of a cerebral infarction); (2) rebleeding of the aneurysm; (3) prior history of stroke; and (4) systemic diseases such as autoimmune diseases, malignancies, liver and/or renal insufficiency, chronic heart disease, chronic lung disease.

Healthy controls were recruited if they fulfilled the following criteria: (1) age ≥ 18 years; (2) no prior history of stroke; and (3) no inflammatory disease or systemic disease. Written informed consent was obtained from all individuals in accordance with the Declaration of Helsinki and under Institutional Review Board approved protocols.

### 2.2. Data Collection

Clinical baseline characteristics together with laboratory and demographic data were collected. The Hunt–Hess grade and modified Fisher (mFisher) grade were employed to assess the initial clinical severity. DCI was identified according to the following criteria: (1) a new focal neurological impairment (such as loss of use, hemiparesis, neglect, aphasia, or hemianopia) or global deficit (a decrease in Glasgow Coma Score of at least 2 points), lasting for more than 1 h; (2) a new cerebral infarction on CT or MR scan that cannot be attributed to other factors (i.e., surgical complication and metabolic derangements) [30]. The assessments were completed by trained clinicians. Functional outcome was evaluated in terms of Glasgow Outcome Scale (GOS) score by in-person or telephone interview at follow-up and was classified as good outcome (GOS 4–5) or poor outcome (GOS 1–3).

### 2.3. Blood Sample Collection and Serum CCL23 Assay

Blood samples of aSAH patients were collected from the cubital vein as soon as possible after admission, within 24 h of symptom onset. The collected samples were immediately transported, centrifuged, aliquoted, and stored at −80 °C until the assay. Serum CCL23 levels were stratified into 3 groups (<8, 8–16, and >16 h) according to the time of blood samples collection, and measured by the enzyme-linked immunosorbent assay (ELISA) Kit for MPIF1 (Cloud-Clone Corp., Wuhan, China) according to the manufacturer’s protocol. All samples were processed by an experienced, blinded laboratory technician.

### 2.4. Statistical Analysis

Data were statistically analyzed using SPSS 26.0 (SPSS Inc., Chicago, IL, USA) and GraphPad Prism 9.0 (GraphPad Software, San Diego, CA, USA) software. Continuous variables were tested for normality distribution and presented as medians and interquartile range (IQR), whilst categorical variables are displayed as numbers and percentages. Continuous and categorical variables were compared using the Mann–Whitney U exact test and χ2/Fisher test, respectively. Correlation analyses between the CCL23 level and the severity of SAH (Hunt–Hess grade and mFisher grade) were performed using Spearman’s correlation coefficient. Univariate logistic regression analysis was used to investigate the risk factors of poor prognosis in aSAH patients, and the independent factors of DCI and poor prognosis were reported as odds ratio (OR) and 95% confidence interval (CI) in multivariate logistic regression analysis. Receiver operating characteristic (ROC) curves were established to evaluated the predictive value of serum CCL23 levels. *p* < 0.05 was considered statistically significant for all analyses.

## 3. Results

### 3.1. Study Population

Of all aSAH patients, 156 patients were initially enrolled, and 54 of them were removed based on the exclusion criteria. The flowchart is detailed in Figure 1. 102 aSAH patients and 61 controls were finally included in the study. The baseline characteristics of patients are shown in Table 1. No significant differences were observed between patients with aSAH and controls in terms of age, sex, body mass index (BMI), and prior medical history.

### 3.2. Changes in Serum CCL23 Levels after aSAH

Serum CCL23 levels were significantly elevated in aSAH patients compared with controls (*p*< 0.001; Figure 2a). To further investigate the changes in serum CCL23 levels over time after aSAH, serum CCL23 levels of aSAH, patients were stratified into three groups (<8 h, 8–16 h, or >16 h) according to the time of sample collection, and the results showed that the temporal profile of serum CCL23 levels exhibited a sustained increase within 24 h after aSAH.

### 3.3. Association of Serum CCL23 Levels with Blood Neutrophils in Inflammation after aSAH

Neuroinflammation is a central component of aSAH [6,31]. The activation and infiltration of neutrophils can express CCL23 and exacerbate brain injury following aSAH [14,17]. Thus, we attempt to ascertain the relationship between serum CCL23 levels and neutrophils in inflammation after aSAH. We found that blood neutrophil level showed the trend of a continuous increase during 24 h after aSAH, which was similar to serum CCL23 levels (Figure 2b). Subsequently, the intimate association between serum CCL23 levels, blood C-reactive protein (CRP) levels, and neutrophil count was observed, as shown in Figure 2c,d.

### 3.4. Correlation between Serum CCL23 Levels with the Initial Clinical Severity after aSAH

We next sought to explore whether serum CCL23 levels were related to initial clinical severity following aSAH. The initial clinical severity of patients was revealed by the Hunt–Hess score and modified Fisher (mFisher) score on admission. As shown in Figure 3a,b, serum CCL23 levels were positively correlated with the Hunt–Hess score and mFisher score, indicating that serum CCL23 levels were intimately related to the initial clinical severity of aSAH.

### 3.5. Serum CCL23 Levels Predicts DCI after aSAH

Subsequently, we investigated the relationship between serum CCL23 levels and DCI. In our study, we found that DCI occurred in 19 of the 102 (18.6%) aSAH patients, and patients with DCI were more likely to have higher serum CCL23 levels than patients without DCI (122.5 pg/mL [IQR 66.4–150.3] vs. 71.6 pg/mL [IQR 48.9–105.3], *p* = 0.003) in this study (Figure 4a and Table 2). Furthermore, serum CCL23 levels, Hunt–Hess grade, cystic aneurysm, and CRP levels were associated with DCI in univariate logistic analysis. Thus, these factors were included in the multivariate logistic regression model and the results showed that serum CCL23 level was an independent predictor of DCI (OR = 1.023 [95%CI 1.002–1.044], *p* = 0.033), while Hunt–Hess score, cystic aneurysm, and CRP levels were not independently associated with DCI. ROC curves were drawn to explore the predictive value of serum CCL23 levels. The area under curve was 0.735 (95%CI 0.610–0.861), indicating its significant predictive value for DCI (Figure 4b).

### 3.6. Serum CCL23 Is an Independent Predictor of Outcome in aSAH

Of all aSAH patients, 42 patients had poor outcomes, while 60 patients showed good outcomes in 6-month follow-up. Patients with good outcomes had increased serum CCL23 levels compared with patients with poor outcomes (102.0 [IQR 70.9–137.1] pg/mL versus 65.1 [IQR 45.8–101.3] pg/mL, *p* < 0.001, Figure 4c). In univariate logistic regression analysis, hypertension, high Hunt–Hess grade, high mFisher grade, large aneurysm size, acute hydrocephalus, DCI, external ventricular drain, elevated CRP levels, high neutrophil count, and elevated serum CCL23 levels are the prognostic factors for 6-month outcomes in aSAH patients (Table 3). When the above significant variables were included in multivariate logistic regression models, serum CCL23 levels were identified as an independent predictor for 6-month poor outcome after aSAH (OR = 1.034 [95%CI 1.007–1.063], *p* = 0.015), while the other factors were not independently associated with 6-month poor outcome. Moreover, Figure 4d shows that serum CCL23 levels could pronouncedly discriminate aSAH patients with poor outcomes in 6-month follow-up (AUC = 0.789, 95%CI 0.662–0.914). Correspondingly, the optimal cutoff for serum CCL23 levels was 76.76 pg/mL, with a specificity of 78.6% and a sensitivity of 70.0%.

## 4. Discussion

To our knowledge, this is the first study to report the relationship between serum CCL23 levels and the prognosis of patients with aSAH. Our data support the vital role of CCL23 in aSAH, based on our research findings: (1) serum CCL23 levels were increased in aSAH patients compared with controls; (2) serum CCL23 was associated with the increase in circulating neutrophils in inflammation after aSAH; (3) CCL23 levels were strongly associated with the initial clinical severity after aSAH; (4) serum CCL23 levels serve as an independent predictor of DCI and poor prognosis in patients with aSAH. Taken together, these findings indicate that serum CCL23 levels could yield insights into the inflammatory response and assist in clinical evaluations of the initial clinical severity, DCI, and poor outcome following aSAH.

As a major mediator responsible for the aggravation of injury and impairments, neuroinflammation occurs in early and delayed brain injury, which are the determining factors for the prognosis of aSAH [6,14]. CCL23 is a chemokine known to regulate the activation and recruitment of immune cells by interacting with CCR1 and subsequently promoting adhesion molecule production [26,32]. Recently, numerous studies have indicated that CCL23 is essential for the inflammation of several human diseases, including ovarian cancer, chronic rhinosinusitis, chronic kidney disease, eosinophilic airway inflammation, and ischemic stroke [18,21,33,34,35,36]. In agreement with these studies, our study showed that aSAH patients had higher CCL23 levels, CRP, and neutrophil count compared with controls, which supports that serum CCL23 was secreted and released to blood in the inflammation after aSAH. Secreted CCL23 may accumulate at the injured brain area, and promote the recruitment of peripheral immune cells [15,36].

Although extensive research has been carried out on CCL23, only a few studies have investigated the temporal profile of serum CCL23 levels within 24 h following disease [19,28,34]. In this study, we demonstrated that the temporal profile of CCL23 exhibited a sustained increase within 24 h after aSAH. Interestingly, we also found that the temporal profile of neutrophil count continuously increased within 24 h following aSAH, which was similar to that of CCL23 levels. Moreover, we observed an increase in neutrophils at an earlier time point than that of serum CCL23 levels. A possible explanation accounting for this phenomenon might be that activated neutrophils infiltrate into the brain parenchyma, and then consistently highly express CCL23 in the inflamed environment, leading to the persistent growth of serum CCL23 levels [17].

In addition, circulating CCL23 has been implicated in the progression of disease [26,28]. Sustained CCL23 secretion modulates the recruitment of other immune cells, which contributes to the rapid amplification of inflammatory response and disease progression [33,35,36]. DCI, a serious complication in the progression of aSAH, is the most important factor that accounts for the high mortality and disability of aSAH [8]. In this study, we found that serum CCL23 levels could significantly predict DCI in patients with aSAH, indicating serum CCL23 also had an impact on the development of DCI. Elevated serum CCL23 may promote the recruitment of peripheral immune cells and the release of pro-inflammatory cytokines, leading to amplification of the inflammatory responses [18]. These inflammatory responses increase the occurrence of DCI after aSAH [11].

The association of elevated serum CCL23 levels with poor outcome is a novel finding. The importance of this result is that it lends support to the potential target of CCL23 for clinical management. Further, our results revealed high discrimination capability of serum CCL23 levels for DCI and poor outcome, which may help clinicians make specific strategies, such as administration of antiplatelet drugs, early open surgery, and aggressive anti-hypertensive therapy, in a timely and aggressive manner [30,37,38].

Our study had several limitations that should be overcome in future studies. First, causal relationships between serum CCL23 levels and aSAH could not be identified due to the cross-sectional design of the study. Secondly, CCL23 concentrations in cerebrospinal fluid (CSF) are remarkably higher than those in peripheral blood, suggesting that CSF CCL23 concentrations might better reflect the inflammation of brain injury [28]. However, CCL23 concentrations in CSF were not measured in our study. This needs to be investigated in the near future. Third, the limited sample size did not allow us to perform any formal statistical validation of these findings, including the CCL23 thresholds. Finally, by interacting with CCR1, circulating CCL23 promotes endothelial cell migration, which may contribute to angiogenesis [39,40]. Thus, circulating CCL23 might play a role in the repair of aSAH. However, this remains speculation at this point and will require further research to clarify.

## 5. Conclusions

In summary, we demonstrate that aSAH patients have significantly higher serum CCL23 levels than controls. Serum CCL23 levels are related to the increase in circulating neutrophils in inflammation after aSAH. Moreover, serum CCL23 levels are associated with initial clinical severity after aSAH. Serum CCL23 levels serve as an independent predictor of DCI and 6-month poor outcome in aSAH patients. Our findings may have important implications for guiding risk stratification and timely decision treatments in aSAH patients.

## Figures and Tables

**Figure 1 jcm-11-06879-f001:**
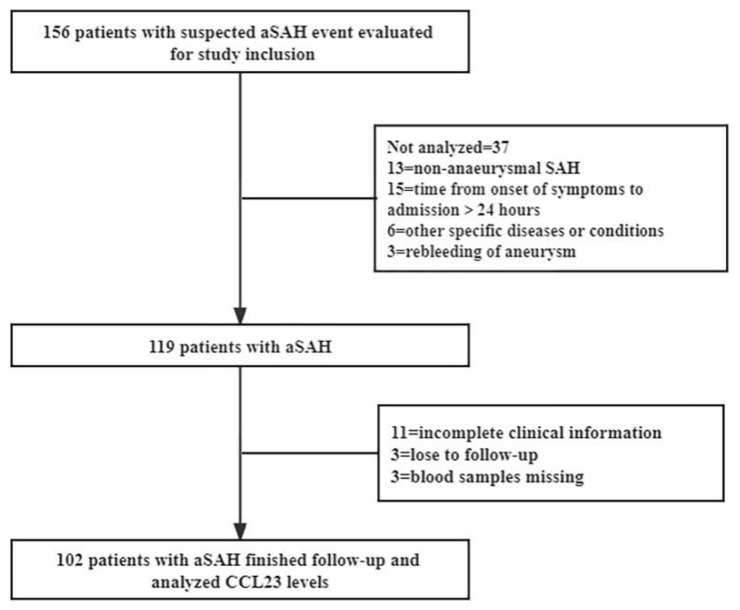
Flow chart of the study population.

**Figure 2 jcm-11-06879-f002:**
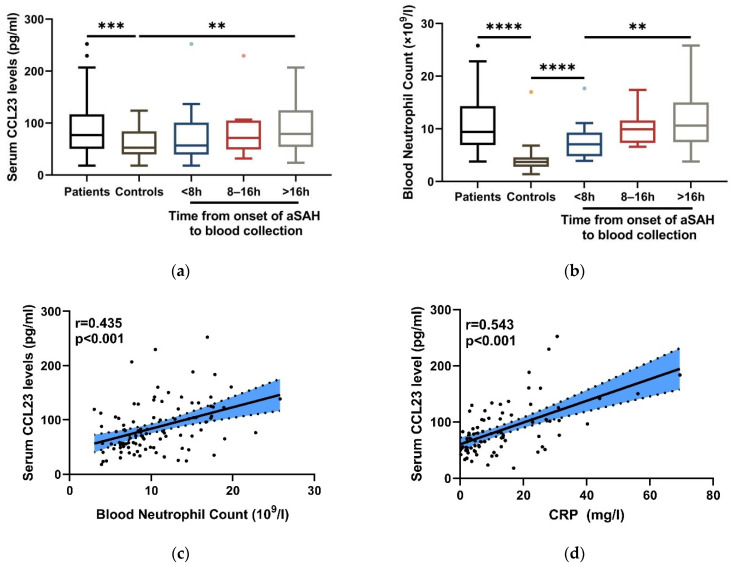
Association between serum CCL23 levels and inflammation after aSAH. (**a**,**b**) Boxplots show the change in serum CCL23 levels and blood neutrophil count in patients with aSAH and in controls. Boxplots represent Turkey, and points outside the whiskers are outliers; ** *p* < 0.01, *** *p* < 0.001, **** *p* < 0.0001. (**c**,**d**) Scatterplot and regression line show correlation between serum CCL23 levels and blood CRP levels, blood neutrophil count at admission. The blue areas represent 95%CI of the regression line; r means Spearmen’s correlation coefficient of two factors.

**Figure 3 jcm-11-06879-f003:**
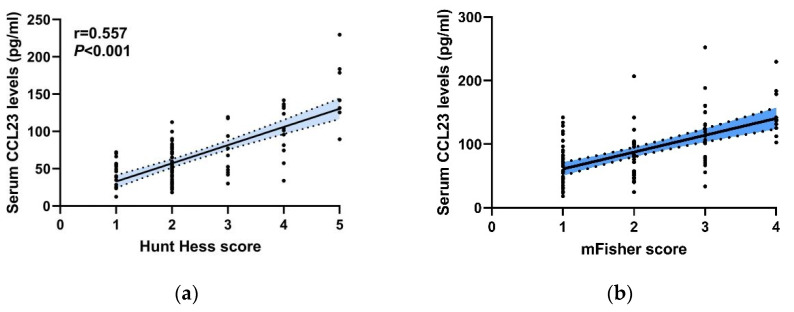
Correlation between serum CCL23 levels with initial clinical severity. (**a**) Scatterplot and regression line show correlation between serum CCL23 levels and Hunt–Hess score at admission. (**b**) Scatterplot and regression line show correlation between serum CCL23 levels and mFisher score at admission. The blue areas represent 95%CI of the regression line; r means Spearmen’s correlation coefficient of two factors.

**Figure 4 jcm-11-06879-f004:**
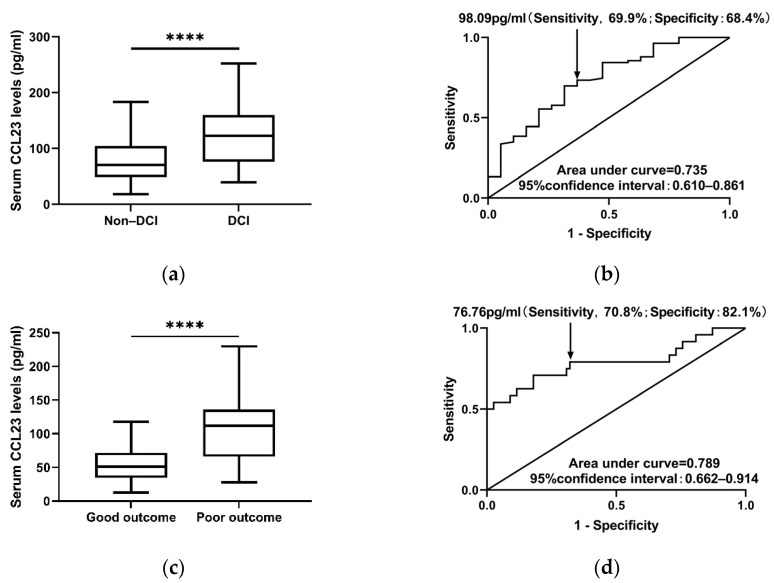
Serum CCL23 levels predict 6-month outcome of aSAH patients. (**a**) Boxplot shows serum CCL23 levels in patients with or without DCI. (**b**) ROC curve shows the discriminatory ability of serum CCL23 levels to predict DCI in aSAH patients. (**c**) Boxplot shows serum CCL23 levels in patients with or without DCI. (**d**) ROC curve shows the discriminatory ability of serum CCL23 levels to predict 6-month poor outcome in aSAH patients. Boxplots represent Turkey boxplots, and points outside the whiskers are outliers. Arrows represent the optimal cutoff value determined using the Youden index. **** *p* < 0.0001.

**Table 1 jcm-11-06879-t001:** Baseline characteristics of the study population.

Characteristics	Patients with aSAH (*n* = 102)	Controls (*n* = 61)	*p* Value
Age, years	55 (46.5–65)	59 (51–66)	0.135
Gender, female	56 (54.9%)	30 (49.2%)	0.616
BMI, kg/m^2^	23.4 (21.6–26.1)	22.3 (20.9–24.7)	0.244
Prior medical history			
Smoker	28 (27.5%)	10 (9.8%)	0.107
Alcohol consumption	21 (20.6%)	11 (18.0%)	0.693
Hypertension	47 (46.1%)	16 (16.4%)	0.012
Diabetes mellitus	9 (8.8%)	4 (6.6%)	0.608
Cardiovascular comorbidities	7 (6.9%)	2 (3.3%)	0.335
Admission status			
Hunt–Hess grade		-	-
1	22 (21.6%)		
2	31 (30.4%)		
3	25 (24.5%)		
4	14 (13.7%)		
5	10 (9.8%)		
Modified Fisher grade		-	-
1	43 (42.2%)		
2	27 (26.5%)		
3	22 (21.6%)		
4	10 (9.8%)		
In-hospital data			
Aneurysmal position, anterior circulation	89 (87.3%)	-	-
Aneurysmal shape, cystic	87 (85.3%)	-	-
Aneurysm size, mm	5.0 (3.7–8.0)	-	-
Aneurysm size > 10 mm	18 (17.6%)	-	-
Surgery, endovascular coiling	50 (49.0%)	-	-
Acute hydrocephalus	26 (25.5%)	-	-
DCI	19 (18.6%)	-	-
External ventricular drain	21 (20.6%)	-	-
Laboratory findings at admission			
CRP, mg/L	8.6 (2.3–21.9)	4.9 (4.3–5.7)	<0.001
Neutrophil count, ×10^9^/L	8.1 (5.6–10.5)	9.4 (6.9–14.3)	<0.001
CCL23, pg/mL	76.8 (50.4–117.0)	52.5 (39.7–84.1)	<0.001

Qualitative data are expressed as numbers (percentages), and quantitative data are expressed as medians (IQRs). BMI, body mass index; CRP, C-reactive protein; DCI, delayed cerebral infarction.

**Table 2 jcm-11-06879-t002:** Univariate logistic regression analysis of characteristics associated with DCI in patients with aSAH.

Characteristics	Patients without DCI (*n* = 83)	Patients with DCI (*n* = 19)	OR	*p*-Value
Age, years	55 (49–64)	55 (43–71)	1.001 (0.960–1.042)	0.981
Gender (female)	38 (45.8%)	8 (42.1%)	0.861 (0.341–2.360)	0.771
BMI, kg/m^2^	23.5 (21.7–26.1)	23.3 (19.9–25.6)	1.097 (0.921–1.307)	0.299
Current smokers	23 (27.7%)	5 (26.3%)	0.872 (0.673–1.130)	0.300
Alcohol consumption	18 (21.7%)	3 (15.8%)	0.677 (0.177–2.583)	0.568
Hypertension	39 (47.0%)	8 (42.1%)	0.821 (0.300–2.247)	0.700
Diabetes mellitus	8 (9.6%)	1 (5.3%)	0.521 (0.061–4.434)	0.551
Cardiovascular comorbidities	4 (4.8%)	3 (15.8%)	3.703 (0.755–18.168)	0.107
Hunt–Hess grade	2 (2–3)	4 (2–4)	1.694 (1.120–2.562)	0.012
Modified Fisher grade	2 (1–3)	2 (2–3)	1.452 (0.900–2.342)	0.126
Aneurysmal position (anterior circulation)	74 (89.2%)	15 (78.9%)	2.193 (0.596–8.061)	0.237
Aneurysmal shape (cystic)	74 (89.2%)	13 (68.4%)	0.264 (0.080–0.866)	0.028
Aneurysm size, mm	5.0 (3.6–8.0)	6.5 (4.5–9.0)	1.034 (0.927–1.152)	0.551
Aneurysm size > 10 mm	15 (18.1%)	3 (15.8%)	0.850 (0.219–3.292)	0.814
Endovascular coiling	40 (48.2%)	10 (52.6%)	1.194 (0.440–3.241)	0.727
Acute hydrocephalus	23 (27.7%)	8 (42.1%)	1.897 (0.677–5.313)	0.223
External ventricular drain	6 (7.2%)	4 (21.1%)	3.422 (0.860–13.614)	0.081
CRP, mg/L	7.8 (3.2–14.1)	22.6 (7.3–28.7)	1.048 (1.006–1.092)	0.025
Neutrophil count, ×109/L	8.8 (6.6–13.7)	11.1 (8.4–16.9)	1.106 (0.996–1.228)	0.061
CCL23, pg/mL	70.6 (48.5–104.4)	122.5 (76.4–160.2)	1.020 (1.009–1.032)	0.001

Qualitative data are expressed as numbers (percentages), and quantitative data are expressed as medians (IQRs). BMI, body mass index; CRP, C-reactive protein; DCI, delayed cerebral infarction.

**Table 3 jcm-11-06879-t003:** Univariate logistic regression analysis of characteristics associated with 6-month poor outcomes.

Characteristics	Good Outcome (*n* = 60)	Poor Outcome (*n* = 42)	OR	*p*-Value
Age, years	54 (46–62)	63 (49.1–69)	1.027 (0.993–1.063)	0.117
Gender (female)	33 (55.0%)	23 (54.8%)	1.010 (0.457–2.230)	0.981
BMI, kg/m^2^	22.8 (21.1–26.0)	23.8 (21.7–26.6)	1.097 (0.921–1.307)	0.299
Current smokers	15 (25.0%)	13 (31.0%)	1.345 (0.559–3.233)	0.508
Alcohol consumption	12 (20.0%)	9 (21.4%)	1.091 (0.413–2.881)	0.861
Hypertension	22 (36.7%)	25 (59.5%)	2.540 (1.131–5.707)	0.024
Diabetes mellitus	4 (6.7%)	5 (11.9%)	1.892 (0.477–7.511)	0.365
Cardiovascular comorbidities	2 (3.3%)	5 (11.9%)	3.919 (0.722–21.257)	0.113
Hunt–Hess grade	2 (1–3)	3 (2–4)	2.663 (1.734–4.089)	<0.001
Modified Fisher grade	1 (1–2)	3 (2–3)	2.521 (1.596–3.983)	<0.001
Aneurysmal position (anterior circulation)	54 (90.0%)	35 (83.3%)	1.800 (0.558–5.802)	0.325
Aneurysmal shape (cystic)	52 (86.7%)	35 (83.3%)	1.300 (0.432–3.910)	0.641
Aneurysm size, mm	5 (3.5–7.0)	6.2 (3.9–10.7)	1.176 (1.044–1.325)	0.008
Aneurysm size > 10 mm	7 (11.7%)	11 (26.2%)	2.687 (0.944–7.648)	0.064
Endovascular coiling	34 (56.7%)	16 (38.1%)	0.471 (0.210–1.053)	0.067
Acute hydrocephalus	12 (20.0%)	19 (45.2%)	3.304 (1.375–7.944)	0.008
DCI	6 (10.0%)	13 (31.0%)	1.027 (0.993–1.063)	0.036
External ventricular drain	2 (3.3%)	8 (19.0%)	6.824 (1.369–34.010)	0.019
CRP, mg/L	5.6 (2.6–13.0)	14.8 (6.7–27.4)	1.084 (1.032–1.140)	0.001
Neutrophil count, ×109/L	8.7 (6.5–13.4)	10.6 (7.5–15.4)	1.084 (0.990–1.186)	0.080
CCL23, pg/mL	61.5 (45.1–86.6)	103.7 (78.0–139.5)	1.024 (1.012–1.036)	<0.001

Qualitative data are expressed as numbers (percentages), and quantitative data are expressed as medians (IQRs). BMI, body mass index; CRP, C-reactive protein; DCI, delayed cerebral infarction.

## Data Availability

The study’s data are available on request from the corresponding author.

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
