# Peer review of "Elevated Serum CCL23 Levels at Admission Predict Delayed Cerebral Ischemia and Functional Outcome after Aneurysmal Subarachnoid Hemorrhage"

_jcm, 2022, doi:10.3390/jcm11236879_

Round 1
Reviewer 1 Report
Lin et al. present a study about the use of elevated serum CCL23 levels at admission to predict delayed cerebral ischemia and functional outcome after aneurysmal subarachnoid hemorrhage. A total of 102 patients with aSAH and 61 control patients were prospectively included in the study and identified that serum CCL23 levels was an independent predictor of DCI and 6-month poor outcome in aSAH patients. CCL 23 seems to be an additional serum biomarker to predict the clinical course of intensive care unit treatment in aSAH patients.
The study is clearly structed and the manuscript is well written. There are only few issues
1. There is an typo within the abstract: ……”: Serum CCl23 levels”…….
2. Citation of the references in the text are not formatted in a uniform way, e.g. ….”mortality and morbidity of aSAH are associated with the biphasic course of SAH (early bleeding and late ischemia)[6]. Initial clinical severity[7] and delayed cerebral ischemia”……Please set al citation at the end of the sentence……
3. Cardiopulmonary complications, e.g. “stunned myocardium”, arrythmia and/or neurogenic edema are well-known early complication in SAH patients. Did the authors have data about a correlation elevated CCL 23 and cardiopulmonary complications?
Author Response
We would like to express our appreciation of the reviewers’ insightful comments concerning our manuscript entitled “Elevated serum CCL23 levels at admission predict delayed cerebral ischemia and functional outcome after aneurysmal subarachnoid hemorrhage”(Manuscript ID: jcm-2012130). The comments have been highly valuable and useful for improving the quality of our paper, as well as important in guiding the direction of our present and future research. The point-by-point responses are listed as follows:
Lin et al. present a study about the use of elevated serum CCL23 levels at admission to predict delayed cerebral ischemia and functional outcome after aneurysmal subarachnoid hemorrhage. A total of 102 patients with aSAH and 61 control patients were prospectively included in the study and identified that serum CCL23 levels was an independent predictor of DCI and 6-month poor outcome in aSAH patients. CCL 23 seems to be an additional serum biomarker to predict the clinical course of intensive care unit treatment in aSAH patients.
The study is clearly structed and the manuscript is well written. There are only few issues
Response to the comment: We are grateful for the reviewer’s positive evaluation.
- There is an typo within the abstract: ……”: Serum CCl23 levels”…….
Response to the comment: Thank you for pointing out this problem. We have checked the abstract of the manuscript and modified the typo in the revised manuscript.
- Citation of the references in the text are not formatted in a uniform way, e.g. ….”mortality and morbidity of aSAH are associated with the biphasic course of SAH (early bleeding and late ischemia)[6]. Initial clinical severity[7] and delayed cerebral ischemia”……Please set al citation at the end of the sentence……
Response to the comment: We appreciate the reviewer for pointing out this fact. We have checked the references thoroughly are now all in a uniform format in the revised manuscript.
- Cardiopulmonary complications, e.g. “stunned myocardium”, arrythmia and/or neurogenic edema are well-known early complication in SAH patients. Did the authors have data about a correlation elevated CCL23 and cardiopulmonary complications?
Response to the comment: Thank you for your instructive suggestion. However, we didn’t collect the data about cardiopulmonary complications at the early stage in aSAH patients. We will take this advice, and explore the correlation between elevated CCL23 and cardiopulmonary complications in the future studies.
Reviewer 2 Report
The authors present an interesting investigation of the relationship between the chemokine CCL23 and deterioration and outcomes after aSAH. Overall the findings are interesting and the research is sounds, but I have a few major concerns and some minor suggestions for edits, as described below:
Overall - Did the authors look at any other inflammatory markers in the blood besides just CCL23, CRP (a very nonspecific marker), and neutrophil count? It would be informative to know whether there is a specific immune profile present in the blood rather than just elevation of one marker. I understand that this project (or at least the portion presented here) was focused on CCL23, but it would be strengthen their argument that CCL23 is uniquely important if they showed this strong relationship was unique or most strongly associated with CCL23 than other cytokines or chemokines.
Minor points:
1. Line 27 - I believe they meant to say that aSAH is an uncommon but life-threatening condition
2. Line 33-34 - The way that this sentences is worded makes the concept confusing. I believe what they meant to say is that the high mortality and morbidity after aSAH is related to bleeding initially and then delayed cerebral ischemia later, but this is not clear from the way it is written.
3. Lines 43-49 – Do the authors mean to say that CCL23 is secreted by the cell types described in the first sentence, or it is expressed on those cell types? In the second sentence, they discuss secreted CCL23 as having an effect on cytokine and adhesion molecule production, but in the first sentence they are presumably discussing cell surface CCL23. Does cell-surface CCL23 have a similar function as secreted CCL23 or a different function? If they are different and they are talking about secreted CCL23 in the rest of the manuscript, then they should discuss which cells produce CCL23 not those that have it on the surface.
4. Line 89-90 – The results section shows blood samples collected at various time points, but the methods section only mentions blood collection at “admission.” It becomes clear in the methods section that the patients are stratified into these groups based on when the blood sample was drawn, but the methods should describe this as some way, such as stating that samples were drawn within 24 hours and patients were stratified into group based on time of the blood sample. It is also unclear if there is a stop point (perhaps 24 hours?) after which it was too late to collect a sample.
5. Figure 2 and Section 2.3 – From the results, it looks like the difference in serum CCL23 levels is driven by the >16hr blood samples, which is different from the timing of increase in blood neutrophil count. Perhaps the difference in neutrophils precedes increases in CCL23 and it would be more informative to look serially throughout the patient’s hospitalization rather than at this time point, which may be too early. Inflammation and immune activation after SAH is a complex and dynamic process, and measures of a marker such as CCL23 at a single and very early time point may not be reflective of the relationship of this marker with clinical features such as DCI that don’t happen for days to a week or two.
6. Lines 168-170 and 194-196 – Were any of the other factors that were significant in the univariate model also predictors of DCI or poor outcome in the multivariate models? This would be important to mention.
7. Lines 232-234 – Similar to the point raised above, based on Figure 2, it looks like neutrophils are actually higher than controls at earlier time points than CCL23 is elevated, so isn’t it also possible that some other factor recruits neutrophils and then they are the major producers of CCL23 once there? The conclusion that CCL23 is responsible for recruitment of neutrophils is difficult to justify given their findings of elevated CCL23 only at the later time points when neutrophils are elevated as early as 6 hours.
Author Response
We would like to express our appreciation of the reviewers’ insightful comments concerning our manuscript entitled “Elevated serum CCL23 levels at admission predict delayed cerebral ischemia and functional outcome after aneurysmal subarachnoid hemorrhage”(Manuscript ID: jcm-2012130). The comments have been highly valuable and useful for improving the quality of our paper, as well as important in guiding the direction of our present and future research. The point-by-point responses are listed as follows:
The authors present an interesting investigation of the relationship between the chemokine CCL23 and deterioration and outcomes after aSAH. Overall the findings are interesting and the research is sounds, but I have a few major concerns and some minor suggestions for edits, as described below:
Overall - Did the authors look at any other inflammatory markers in the blood besides just CCL23, CRP (a very nonspecific marker), and neutrophil count? It would be informative to know whether there is a specific immune profile present in the blood rather than just elevation of one marker. I understand that this project (or at least the portion presented here) was focused on CCL23, but it would be strengthen their argument that CCL23 is uniquely important if they showed this strong relationship was unique or most strongly associated with CCL23 than other cytokines or chemokines.
Response to the comment: We thank the reviewer for the generally positive assessment of the contents of our manuscript, and for the constructive suggestions.While it is an intersting and meaningful research direction to find a specific immune profile in the peripheral blood after aSAH, we unfortunately didn't explore this in our study. We will explore this in the future studies.
- Line 27 - I believe they meant to say that aSAH is an uncommon but life-threatening condition
Response to the comment: We appreciate the reviewer for pointing this out. We have modified the sentence in the revised manuscript (line 27).
- Line 33-34 - The way that this sentences is worded makes the concept confusing. I believe what they meant to say is that the high mortality and morbidity after aSAH is related to bleeding initially and then delayed cerebral ischemia later, but this is not clear from the way it is written.
Response to the comment: We thank the reviewer for this suggestion. We have modified this sentence with a more clear expression in the revised manuscript (line 33-34). As follows: The initial bleeding and delayed ischemia following aSAH have a major impact on the high disability rate and mortality of aSAH
3.Lines 43-49 – Do the authors mean to say that CCL23 is secreted by the cell types described in the first sentence, or it is expressed on those cell types? In the second sentence, they discuss secreted CCL23 as having an effect on cytokine and adhesion molecule production, but in the first sentence they are presumably discussing cell surface CCL23. Does cell-surface CCL23 have a similar function as secreted CCL23 or a different function? If they are different and they are talking about secreted CCL23 in the rest of the manuscript, then they should discuss which cells produce CCL23 not those that have it on the surface.
Response to the comment: We thank the reviewer for this professional opinion. Actually, what we want to say is that CCL23 is secreted by several immune cells. We have modified our expression in the revised manuscript (line 44-44). As follows: CC chemokine ligand 23 (CCL23), a member of the CC chemokine family, is secreted by several immune cells (e.g. neutrophils, monocytes, eosinophils, dendritic cells, and macrophages).
4.Line 89-90 – The results section shows blood samples collected at various time points, but the methods section only mentions blood collection at “admission.” It becomes clear in the methods section that the patients are stratified into these groups based on when the blood sample was drawn, but the methods should describe this as some way, such as stating that samples were drawn within 24 hours and patients were stratified into group based on time of the blood sample. It is also unclear if there is a stop point (perhaps 24 hours?) after which it was too late to collect a sample.
Response to the comment: We thank the reviewer for pointing out this problem. Our blood samples were collected as soon as possible after admission, within 24 h of symptom onset. Moreover, the grouping detail has been described. These were written in the revised manuscript (line 89-93).
5.Figure 2 and Section 2.3 – From the results, it looks like the difference in serum CCL23 levels is driven by the >16hr blood samples, which is different from the timing of increase in blood neutrophil count. Perhaps the difference in neutrophils precedes increases in CCL23 and it would be more informative to look serially throughout the patient’s hospitalization rather than at this time point, which may be too early. Inflammation and immune activation after SAH is a complex and dynamic process, and measures of a marker such as CCL23 at a single and very early time point may not be reflective of the relationship of this marker with clinical features such as DCI that don’t happen for days to a week or two.
Response to the comment: Thank you for your rigorous consideration. In our opinion, serum CCL23 at very early time point could predict DCI. The reasons are as follows:
Among various factors that influence the development of DCI, early brain injury (EBI) is thought to play an important role. Chemokines are the critical responses in the inflammatory cascade. The enhancement of serum CCL23 levels at very early time point may promotes the recruitment of peripheral immune cells and the amplification the inflammatory responses, leading to aggravation of EBI. Moreover, serum CCL23 levels were related to direct injury caused by initial bleeding (reflected by the initial clinical severity) in our study. Direct injury caused by initial bleeding is the important part of EBI. Thus, serum CCL23 levels at very early time point was closely related to EBI after aSAH, and we believe that CCL23 could predict DCI. However, there is no direct evidence for these explanations, and further investigation is needed. Furthermore, blood samples over 24 h would be collected in future studies for better explore the relationship between serum CCL23 and aSAH.
6.Lines 168-170 and 194-196 – Were any of the other factors that were significant in the univariate model also predictors of DCI or poor outcome in the multivariate models? This would be important to mention.
Response to the comment: We agree with the reviewer that whether the other factors could serve as predictors of DCI or poor outcome in the multivariate models should be mentioned. We have added the description “while Hunt-Hess score, cystic aneurysm, and CRP levels were dependent variables of DCI.” in line 170-171, and “while the other factors were dependent variables of 6-month poor outcome.” in line 198-199 in revised manuscript.
- Lines 232-234 – Similar to the point raised above, based on Figure 2, it looks like neutrophils are actually higher than controls at earlier time points than CCL23 is elevated, so isn’t it also possible that some other factor recruits neutrophils and then they are the major producers of CCL23 once there? The conclusion that CCL23 is responsible for recruitment of neutrophils is difficult to justify given their findings of elevated CCL23 only at the later time points when neutrophils are elevated as early as 6 hours.
Response to the comment: We thank the reviewer for noticing that neutrophils increased at earlier time points than serum CCL23, and we share the reviewer’s opinion on the unpersuasive explanations “activated neutrophils migrate into the injury area in response to CCL23 gradients”. We have modified our statement in the revised discussion(line 235-241), as follow: Interestingly, we found that the temporal profile of neutrophil count was also continuous increased within 24 h following aSAH, which was similar to that of CCL23 levels. Moreover, we observed an increase in neutrophils at an earlier time point than that of serum CCL23 levels. A possible explanation account for this phenomenon might be that activated neutrophils infiltrate into the brain parenchyma, and then consistently highly express CCL23 in the inflamed environment, leading to the persistently growth of serum CCL23 levels
Round 2
Reviewer 2 Report
I appreciate the authors thoughtful responses to my prior comments, and appropriate edits. I have outlined a few additional comments below.
Lines 44-45 - I think it is very difficult to say that inflammation is the major driver of initial clinical severity (this is most likely primarily due to the bleeding itself, which likely causes an inflammatory cascade). When the authors changed the wording in the paragraph above to say initial clinical severity instead of early brain injury, they also changed it in this section. These are 2 separate things. The initial clinical severity is related to the bleed itself and potential other factors, and (as described in the paper they reference) early brain injury occurs in the first 72 hours. I think it is reasonable to say that inflammation plays a role in early brain injury, but to say it plays a role in initial severity (usually defined by initial clinical exam and imaging immediately after the bleed) is confusing.
Lines 177 and 205 - I believe the authors are using incorrect statistical terminology here. To say something is a "dependent variable" is very different from that variable not having an independent association with outcome. Since they included all of the variables they mentioned in the multivariable model, if CCL23 is the only variable that is an independent predictor for DCI and outcome, then they should state that the other variables are "not independently associated with DCI/outcome" not that they are dependent variables.
Author Response
We are very grateful for your careful reading of my responses and considered comments, which help us to correct some mistakes in our revision. The following is our point-by-point response to the reviewers' comments
I appreciate the authors thoughtful responses to my prior comments, and appropriate edits. I have outlined a few additional comments below.
Response to the comment: Again, we thanks for your valuable comments.
Lines 44-45 - I think it is very difficult to say that inflammation is the major driver of initial clinical severity (this is most likely primarily due to the bleeding itself, which likely causes an inflammatory cascade). When the authors changed the wording in the paragraph above to say initial clinical severity instead of early brain injury, they also changed it in this section. These are 2 separate things. The initial clinical severity is related to the bleed itself and potential other factors, and (as described in the paper they reference) early brain injury occurs in the first 72 hours. I think it is reasonable to say that inflammation plays a role in early brain injury, but to say it plays a role in initial severity (usually defined by initial clinical exam and imaging immediately after the bleed) is confusing.
Response to the comment: We apologize for the inappropriate modifications in the revised manuscript. We have modified the sentence in line 43-44, as follows: Immune reaction is increasingly recognized to have a pronounced impact on early brain injury and DCI following aSAH.
Lines 177 and 205 - I believe the authors are using incorrect statistical terminology here. To say something is a "dependent variable" is very different from that variable not having an independent association with outcome. Since they included all of the variables they mentioned in the multivariable model, if CCL23 is the only variable that is an independent predictor for DCI and outcome, then they should state that the other variables are "not independently associated with DCI/outcome" not that they are dependent variables.
Response to the comment: Thank you for your thoughtful suggestions. We agree with the reviewer’s opinion that "not independently associated with DCI/outcome" is the appropriate expression rather than "dependent variable". We have modified the sentences in line 171-172 and line 199-200, as follows:
"while Hunt-Hess score, cystic aneurysm, and CRP levels were not independently associated with DCI.", and "while the other factors were not independently associated with 6-month poor outcome."